# Acid Ceramidase Deficiency: Bridging Gaps between Clinical Presentation, Mouse Models, and Future Therapeutic Interventions

**DOI:** 10.3390/biom13020274

**Published:** 2023-02-01

**Authors:** Annie Kleynerman, Jitka Rybova, Mary L. Faber, William M. McKillop, Thierry Levade, Jeffrey A. Medin

**Affiliations:** 1Department of Pediatrics, Medical College of Wisconsin, Milwaukee, WI 53226, USA; 2Laboratoire de Biochimie Métabolique, CHU Toulouse, and INSERM U1037, Centre de Recherches en Cancérologie de Toulouse, Université Paul Sabatier, 31062 Toulouse, France; 3Department of Biochemistry, Medical College of Wisconsin, Milwaukee, WI 53226, USA

**Keywords:** ceramide, lysosomal storage disorder, SMA-PME, Farber disease, lipogranulomatosis, acid ceramidase

## Abstract

Farber disease (FD) and spinal muscular atrophy with progressive myoclonic epilepsy (SMA-PME) are ultra-rare, autosomal-recessive, acid ceramidase (ACDase) deficiency disorders caused by *ASAH1* gene mutations. Currently, 73 different mutations in the *ASAH1* gene have been described in humans. These mutations lead to reduced ACDase activity and ceramide (Cer) accumulation in many tissues. Presenting as divergent clinical phenotypes, the symptoms of FD vary depending on central nervous system (CNS) involvement and severity. Classic signs of FD include, but are not limited to, a hoarse voice, distended joints, and lipogranulomas found subcutaneously and in other tissues. Patients with SMA-PME lack the most prominent clinical signs seen in FD. Instead, they demonstrate muscle weakness, tremors, and myoclonic epilepsy. Several ACDase-deficient mouse models have been developed to help elucidate the complex consequences of Cer accumulation. In this review, we compare clinical reports on FD patients and experimental descriptions of ACDase-deficient mouse models. We also discuss clinical presentations, potential therapeutic strategies, and future directions for the study of FD and SMA-PME.

## 1. Introduction

### 1.1. Acid Ceramidase and Metabolites

Encoded by the *ASAH1* gene, acid ceramidase (ACDase) is a lysosomal hydrolase involved in cellular ceramide (Cer) homeostasis [1]. The enzyme was first identified as a catalyst for amide bond hydrolysis involving Cers in rat brains by Gatt in 1963 [2]. In 1995, following large-scale purification from human urine, ACDase was defined as a heterodimer with an α (13 kDa) and a β (40 kDa) subunit [1]. The enzyme is synthesized as a single 53–55 kDa precursor polypeptide that is post-transcriptionally modified in lysosomes into the α and β subunits [3]. Enzymatic cleavage into these subunits is crucial for ACDase activity [3]. In 2018, ACDase’s crystal structure was described for both the proenzyme and cleaved forms [4]. A hydrophobic channel leading to the active site is revealed during the autocleavage process [4]. An optimal pH of 4.5–5 for enzymatic activity and the presence of a mannose-6-phosphate tag suggest a likely role for ACDase in lysosomal function [5]. ACDase cleaves Cers into sphingosine and free fatty acids. The enzyme has a higher catalytic efficiency for C12 Cers compared to other Cers [6]. ACDase is reported to also have reverse synthesis abilities at a lysosomal pH near 6.0, preferentially using C12:0 or C14:0 fatty acids and natural D-erythro sphingosine isomers to generate Cer [5]. This underscores the importance of ACDase’s role in the complex regulation of Cer and sphingosine metabolism [5]. Lately, it has been shown that ACDase also acts with fatty acid amide hydrolase and N-acylethanolamine-hydrolyzing acid amidase to degrade N-acylethanolamine (NAE) to ethanolamine and free fatty acids at a pH of 4.5, with a possible preference for NAEs over Cers, though these molecules are present at a much lower abundance in cells [7].

Cer, ceramide-1-phosphate (C1P), sphingosine, sphingosine-1-phosphate (S1P), and NAEs are all bioactive lipids involved in cellular signaling, exerting a range of biological actions via various receptors [8]. Each Cer species is characterized by its saturation, hydroxylation, fatty acid chain length, and sphingoid base moiety. Cers have a central role in the formation and catabolism of complex sphingolipids. Moreover, Cers regulate antiproliferative processes such as growth inhibition, apoptosis, differentiation, and senescence [8,9,10,11]. Cers have also been revealed to be essential cell cycle modulators, regulating morphological transformations and checkpoints, binding to transcription factors, and altering mitotic spindle assembly [12]. The phosphorylated form of sphingosine, S1P, is important for cell survival, cell motility, angiogenesis, immunity, and inflammation [9]. NAEs such as oleoylethanolamide, anandamide, and palmitoylethanolamide have anorexic, cannabimimetic, and anti-inflammatory functions, respectively [7]. ACDase’s involvement in the production of these key bioactive lipids highlights the extensive intricacy of the enzyme’s mechanism and downstream effects.

### 1.2. Farber Disease

Mutations in the *ASAH1* gene can result in Farber disease (FD) or spinal muscular atrophy with progressive myoclonic epilepsy (SMA-PME). Both are ultra-rare lysosomal storage disorders (LSDs) that are poorly understood and often misdiagnosed. In 1947, Dr. Sidney Farber described the first case of a 14-month-old infant with “disseminated lipogranulomatosis” and, in 1952, published a case series of three patients [13]. Since then, more than 200 cases of ACDase deficiency have been published. Based on varying clinical manifestations, FD is categorized into seven subtypes [14]. Type 1, or classical FD, is the most prevalent. Usually not surviving beyond early childhood, type 1 patients commonly present with a lung disease, intellectual disability, hoarse voice, lipogranulomas, and painful, swollen joints. Types 2 and 3 include cases with longer survival due to attenuated neurological involvement. Type 4 FD, or neonatal visceral, is defined by severe hepatosplenomegaly in the neonatal period, resulting in death. Type 5 FD patients’ most prominent feature is progressive neurological decline and seizures, while type 6 includes signs of both FD and Sandhoff disease, a disorder resulting from hexosaminidase A and B enzyme deficiency. Lastly, type 7 is defined by prosaposin deficiency.

Recently, we have characterized FD and SMA-PME as a spectrum of disorders rather than distinct subgroups [15,16]. In 2018, Yu et al. described the broad range of system-wide signs and symptoms that encompass ACDase deficiency [15]. Differing phenotypic presentations such as variable ACDase activity, the presence/absence of subcutaneous nodules, and altered lifespans were observed in siblings with FD, individuals with the same mutations, and patients diagnosed at different ages [15]. Since then, more unique cases have been published with overlapping FD and SMA-PME phenotypes [17,18]. In a recent case, a patient with lifelong SMA began developing symptoms at age 45 like those seen in FD, including a hoarse voice, subcutaneous nodules, and atrophy of the tongue [17]. Sanger and next-generation sequencing revealed several compound heterozygous variants in the *ASAH1* gene that had previously been described in FD patients, confirming the first documented case of SMA progressing into FD [17]. Cases such as this one with SMA progressing to FD or others in which FD and SMA-PME coexist in patients further support the concept that FD and SMA-PME fall on a spectrum of ACDase deficiency instead of being two separate disorders [17,18]. 

### 1.3. Distinguishing SMA, PME, and SMA-PME

First discovered in the early 1890s by Werdnig and Hoffman, spinal muscular atrophy (SMA) is a clinically and genetically heterogeneous group of disorders characterized by progressive muscle weakness and paralysis due to spinal cord α motor neuron degeneration [19,20]. Mutations in the survival motor neuron 1 (*SMN1*) gene cause SMA and are most often inherited as an autosomal recessive trait [19,21]. SMA encompasses four subtypes (SMAI, SMAII, SMAIII, and SMAIV) based on severity, age of onset, and degree of motor function [19]. With SMAI being the most severe and SMAIV the least severe, symptoms can include absent tendon reflexes, decreased muscle tone, reduced limb movements, impaired swallowing and eating, difficulties breathing, and fasciculations [19]. Prognosis depends on the type of SMA and when treatment is initiated. Some patients die in childhood despite treatment efforts, while others can remain stable through adulthood with or without treatment [19,22]. 

Progressive myoclonus epilepsy (PME) is described as a group of progressive epileptic disorders characterized by seizures and muscle contractions [23]. Unverricht–Lundborg disease, a type of PME, is caused by mutations in the *CSTB* gene [23]. In another type of PME, Lafora disease, the *EPM2A* gene or *NHLRC1* gene contains PME-causing mutations [23]. Other types of PME are more secondary in nature such as in Batten disease and mitochondrial myopathies [23]. Clinical signs in all types of PME may include: twitching, jerking, impaired cognition, poor vision, decreased hearing ability, impaired motor coordination, gastrointestinal discomfort, bladder dysfunction, and thyroid complications [23]. The underlying cause of the disorder dictates the treatment plan and prognosis in patients with PME [23]. 

A novel subgroup of ACDase deficiency, SMA-PME, does not present with classical signs or symptoms of FD, adding a new challenge to diagnosis and treatment of this disorder [16]. First described by Jankovic as “hereditary myoclonus and progressive muscular atrophy” in 1978, SMA-PME was identified in patients with progressive muscle wasting and weakness who later developed myoclonus and limb jerking [24]. Further analysis of several patients with SMA-PME identified the mutations responsible within the *ASAH1* gene [21]. SMA-PME-related muscle weakness is often seen in children between the ages of 2 and 7; yet several cases have also been seen in adolescents up to the age of 15 [25,26]. Development of lower motor neuron disease is often the first clinical sign of SMA-PME [27]. Progressive seizures, difficulty walking, tremors, myoclonus, falling, and cognitive decline all progress after the first appearance of symptoms [21,28,29,30]. Sensorineural hearing loss has also been reported in several patients [30]. Respiratory complications and/or refractory seizures often cause early adulthood mortality in SMA-PME patients [30].

### 1.4. ACDase Deficiency Mouse Models

The low number of FD and SMA-PME case studies, a poor disease prognosis, and limited therapeutic options have made the development of an ACDase-deficient mouse models crucial for understanding these ultra-rare disorders and developing treatments. The first two published attempts at *Asah1* knockout mouse model generation resulted in early embryonic lethality and infertility [6,31,32]. In the first, Li et al. introduced a large insertion with a pmACgKO vector into intron 12 of the endogenous mouse *Asah1* gene, leading to an embryonic lethal phenotype in homozygous mice [32]. The breeding of heterozygous mice resulted in no *Asah1*^−/−^ individuals from over 150 offspring, while the ratio of wild-type to *Asah1*^+/−^ individuals was 1:2. A subsequent study from this group found that the transgenic homozygous embryos died at the two-cell stage [33]. These investigators found that *Asah1*^+/−^ mice survived with normal lifespans but developed a progressive lipid storage disorder in skin, lung, liver, and bone [32]. These results were correlated with an up-to-two-fold increase in the Cer content in these tissues and a reduction in ACDase activity [32]. Yet, given the autosomal recessive inheritance of ACDase deficiency, the significant difference in Cer content between *Asah1*^+/−^ mice and controls was an unanticipated finding. 

In the second model, conditional *Asah1* knockout mice were generated by Eliyahu et al. [31]. Using tamoxifen (TM) to induce Cre recombinase nuclear translocation with a LoxP-flanked (floxed) *Asah1* gene construct, homozygous floxed offspring were bred to Cre transgenic mice to generate ACDase knockout animals (cACKO) that were homozygous for the floxed ACDase allele and heterozygous for the Cre allele [31]. When the *Asah1* gene was knocked out by treatment with TM, ACDase decreased in ovaries, and Cer content correspondingly increased [31]. The cACKO mice also displayed infertility, suggesting an important role for ACDase in reproduction [31]. This mouse model was later used to study ACDase’s role in hepatic fibrosis and non-alcoholic fatty liver disease [31,34].

In 2013, we generated the first viable mouse model of ACDase deficiency through “knock-in” of a recognized human *ASAH1* patient mutation, proline (P) 362 to arginine (R), into the murine *Asah1* gene locus (P361R) [35]. The homozygous mice (*Asah1*^P361R/P361R^) experienced growth retardation starting at around 3 weeks of age, progressively losing weight and eventually dying around 7–13 weeks of age. With hallmark features, including systemic Cer accumulation and the buildup of foamy macrophages in organs, this model has proven to be useful for studying pathology relevant to patients with FD. To date, a variety of organ systems have been studied in the *Asah1*^P361R/P361R^ model, including the hematopoietic system, central nervous system (CNS), eyes, lungs, skin, and liver [35,36,37,38,39,40,41,42,43]. 

Another viable mouse model of ACDase deficiency was described by Beckmann et al. in 2018 [44]. Their *Asah1^tmEx1^* mouse was generated by a targeted deletion in exon 1 of *Asah1*, which encodes the signal sequence. This deletion resulted in the formation of a variant ACDase protein from an alternative start codon that lacked lysosomal targeting. Similar to *Asah1*^P361R/P361R^ mice, *Asah1^tmEx1^* mice weighed significantly less than wild-type animals at the age of weaning, progressively lost weight from week 5 onwards, and ultimately died around 6–7 weeks of age. *Asah1^tmEx1^* mice exhibited disruption of ACDase enzyme activity in the thymus, liver, spleen, and bone marrow. Cer and sphingomyelin accumulation was present in the lungs, liver, spleen, kidney, muscle, and brain of *Asah1^tmEx1^* mice. Histiocytic infiltration in lymphoid organs, lung inflammation, liver fibrosis, muscular disease, and mild kidney injury were also found in *Asah1^tmEx1^* mice [44]. Using the *Asah1^tmEx1^* model, a follow-up study by Beckmann et al. investigated acid sphingomyelinase (Asm), a lysosomal enzyme upstream of ACDase that synthesizes Cers, as a potential therapeutic target for FD [45]. This study generated a double ACDase- and Asm-deficient mouse, which displayed prolonged survival, fewer disease manifestations, and decreased Cer accumulation [45].

To investigate the effects of endothelium-specific *Asah1* gene deletion, Yuan et al. developed *Asah1*^fl/fl^/EC^cre^ mice by crossing *Asah1^tmEx1^* floxed mice with Tek-Cre mice (Tie2-Cre transgenic mice from the Jackson Laboratory) [46]. Endothelial ACDase was found to regulate hyperglycemia-induced NOD-, LRR-, and pyrin-domain-containing protein 3 (NLRP3) inflammasome activation, a novel mechanism of diabetic vascular endothelial dysfunction [46]. Importantly, this study suggested that lysosomal ACDase regulates endothelial exosome release, which, in turn, mediates the secretion of NLRP3 inflammasome products [46]. In a separate study conducted by the Li group, a smooth muscle-specific *Asah1* gene knockout was investigated [47]. *Asah1^tmEx1^* floxed mice were crossed with SM22α-Cre transgenic mice (SM^Cre^ mice from the Jackson Laboratory) to generate *Asah1*^fl/fl^/SM^Cre^ mice [47]. These smooth muscle-specific *Asah1* gene knockout mice were used to study the high mobility group box 1 (HMGB1) signaling axis in ACDase-deficiency-mediated vasculopathy [47]. Induced by ACDase deficiency and Cer accumulation, HMGB1 was suggested to promote smooth muscle cell (SMC) migration and proliferation in coronary arterial myocytes (CAMs) isolated from smooth muscle-specific ACDase knockout mice [47]. The endothelium-specific and smooth muscle-specific *Asah1* gene knockout mouse models provide novel insights into the pathogenesis of metabolic-disorder-related vascular remodeling [46,47].

In 2020, Li et al. generated a knockout mouse strain (*Asah1*^fl/fl^/Podo^Cre^) with a podocyte-specific deletion of the α subunit of ACDase by crossing *Asah1* floxed mice with podocyte-specific Cre recombinase (Podo^Cre^) mice from the Jackson Laboratory [48]. *Asah1*^fl/fl^/Podo^Cre^ mice were found to have podocytopathy and evidence of nephrotic syndrome, highlighting ACDase’s role in podocyte development [48]. A downstream experiment in which the Asm protein encoding gene *Smpd1* was further knocked out revealed the protective effects of Asm ablation [48]. Double-knockout *Smpd1*^−/−^/*Asah1*^fl/fl^/Podo^Cre^ mice had significantly less Cer accumulation in the glomerulus and decreased podocyte injury, further supporting Asm’s role as a therapeutic target in cases of ACDase deficiency [48]. 

To study ACDase’s role in the inflammatory infiltrate found in ulcerative colitis, Obeid et al. developed a myeloid-specific ACDase conditional knockout mouse model, Mye AC cKO, by crossing the floxed *Asah1* mice from Eliyahu’s study with strain B6.129P2-Lyz2tm1(cre)Ifo/J [31,49]. In the same study, Obeid’s group also bred *Asah1* floxed mice with strain B6.SJL-Tg(Vil-cre)997Gum/J to create an intestinal epithelial ACDase conditional knockout mouse model (Vil AC cKO) [31,49]. They reported upregulated ACDase expression in the intestinal inflammatory infiltrate associated with ulcerative colitis [49]. Clinical parameters such as weight loss, colon shortening, and anemia were attenuated with ACDase knockout animals in dextran sulfate sodium-induced colitis [49]. Importantly, Mye AC cKO mice displayed altered neutrophil recruitment, inflammation, and intestinal tumorigenesis [49].

In 2012, the Melki group published a zebrafish model in which they performed a morpholino-based knockdown of fish *asah1b*, the fish ortholog of human *ASAH1*. This knockdown led to loss of motor neuron axonal branching, reflective of the increased apoptosis in the spinal cord that has been found in SMA-PME [21]. No mouse models of SMA-PME have been published to date. That said, we have generated a unique SMA-PME model by selective breeding of our previously described mouse model of FD (manuscript submitted).

Primary findings across all mouse models of ACDase deficiency include Cer accumulation, macrophage infiltration, presence of Farber body inclusions in many tissues, and elevation of plasma monocyte chemoattractant protein-1 (MCP-1) concentration. In this review, we examine these mouse models of ACDase deficiency and compare findings with patient cases throughout the published clinical history of FD and SMA-PME.

## 2. Disease Manifestations Due to Acid Ceramidase Dysfunction

### 2.1. Pulmonary

Pulmonary complications are common in cases of both FD and SMA-PME, causing fatal complications. Clinical signs of FD include aphonia, expiratory stridor, sternal retraction, and wheezing or labored breathing upon auscultation [50,51,52,53,54]. Laryngeal and upper airway nodule formation causes extreme respiratory distress, often requiring tracheostomy as a lifesaving procedure [54,55,56]. Chest X-rays detect nodular opacities, hyperinflation, consolidation, and atelectasis [53,54,57,58]. In one case of FD, bronchoscopy uncovered a circular contraction in the laryngeal subglottic region [59]. Upon post-mortem examination and bronchial alveolar lavage, significant cellular infiltration and inflammation characterized by enlarged, lipid-filled macrophages were observed [50,60]. Using chest CT to image and Hounsfield units to quantify, a case study presented adipose tissue in the lungs of a patient with pure SMA evolving into FD [17]. Another study described the lung tissue of a FD patient as poorly expanded with increased peribronchial and perivascular tissue present [61]. Further structural investigation revealed lung histiocytes containing curvilinear storage bodies (characteristic Farber bodies) [61]. In SMA-PME, the motor neuron disease’s severe effects on respiratory muscles often lead to respiratory failure [21,25,26,62]. These findings highlight the importance of incorporating respiratory screening into the initial exam, as pulmonary pathology remains the primary cause of mortality in patients with ACDase deficiency.

In 2018, we investigated pulmonary dysfunction and chronic lung injury in our mouse model of ACDase deficiency, *Asah1*^P361R/P361R^. We uncovered several parallels between the human disease condition and the *Asah1*^P361R/P361R^ mice in this context (Figure 1) [39]. Specifically, altered lung mechanics in the mice, such as low lung compliance and increased airway resistance, were seen, indicating considerable overlap with descriptions of FD patients’ “poorly expanded” lungs [39,61]. In both the post-lavage lung tissue and in bronchial alveolar lavage fluid, all analyzed Cers (C16:0, C18:0, C20:0, C22:0, C24:1, C24:0, C26:0) were elevated. Significant inflammation, cellular infiltration, edema, and increased vascular permeability contributed to the impaired lung function observed in the *Asah1*^P361R/P361R^ mice [39]. Beckmann et al. described neutrophil infiltration and elevated levels of myeloid-derived suppressor cells in lung sections of their *Asah1^tmEx1^* mice [44]. Dendritic cell and B cell numbers per lung were significantly decreased, while T cell numbers per lung were not found to be statistically different [44]. Notably, macrophage numbers were unchanged in comparison to wild-type mice. Given the involvement of macrophage-driven inflammation in ACDase-deficiency-related pulmonary pathology, the lack of significant lung macrophage recruitment is important to note when using *Asah1^tmEx1^* mice as a pre-clinical therapeutic model [44,50,60]. The *Asah1^tmEx1^* model provides insight into cellular trends pertaining to ACDase deficiency, while the *Asah1*^P361R/P361R^ model exhibits a gross phenotype that parallels the human condition, rendering both models important pre-clinical tools for investigating respiratory therapeutics pertaining to FD and SMA-PME. 

### 2.2. Hepatic

Hepatomegaly, and a palpable liver upon medical examination, paired with liver enzymes elevated over three-fold is common in cases of classic FD [50,57,60,63,64,65]. In one of Dr. Sidney Farber’s first case reports, he noted hepatic necrosis, faint cytoplasmic staining within hepatocytes and Kupffer cells, and dilation of liver sinuses [50]. A subsequent ultrastructural study in 1986 described curvilinear tubular bodies unique to FD and zebra bodies in hepatocytes, Kupffer cells, and endothelial cells in a liver biopsy from a 1-week-old boy with FD [66]. Since then, severe liver pathology has been reported as a cause of early death in several FD patients, particularly affecting those at the severe end of the spectrum [16]. In these neonatal visceral cases, a rarer phenotype of extensive hepatomegaly and histiocytosis is associated with death. These patients often die before the age of 1, making diagnosis possible only through enzyme, gene, and biochemical analyses [16]. In several case studies, such infants presented with liver fibrosis, ascites, and cholestatic jaundice [55,67,68]. In another case of FD, a 6-month-old infant initially diagnosed with neonatal giant cell hepatitis underwent liver transplantation [55]. The proper diagnosis of FD was made 3 months after transplantation due to the increasing appearance of subcutaneous nodules [55]. Retrospective histological analysis of the pre-transplantation liver biopsy demonstrated characteristic Farber bodies, further supporting the diagnosis of FD [55]. In 2008, Willis et al. reported highly vacuolated histiocytes, cholestasis, foamy cells, and hepatic necrosis in two infants from the same family with type 4 FD [67]. Interestingly, only one of the siblings displayed the characteristic subcutaneous nodules and Farber bodies in the liver biopsy [67]. 

Analogous to the clinical findings characterized in neonatal visceral FD, our *Asah1*^P361R/P361R^ mouse exhibited hepatomegaly, elevated liver enzymes, progressive fibrosis, inflammatory cell infiltration, and cell death (Figure 1) [42,65]. All Cers measured (C16:0, C18:0, C20:0, C22:0, C24:1, C24:0) were increased in *Asah1*^P361R/P361R^ mouse livers compared to the livers of the controls [42]. As in the *Asah1*^P361R/P361R^ mouse, fibrosis and foamy macrophage infiltration were detected in the livers of *Asah1^tmEx^* mice [42,44]. Hepatocellular injury markers, alanine aminotransferase and aspartate aminotransferase, also increased as the *Asah1^tmEx1^* mice aged [44]. Interestingly, total serum protein, albumin levels, and bilirubin levels were not statistically different in *Asah1^tmEx1^* mice when compared to those of wild-type littermates [44]. 

The limited number of cases describing the effects of ACDase deficiency on liver function in FD patients and the rapid progression of the neonatal visceral subtype make mouse models such as these particularly important for the development of therapeutic interventions for this disorder. Further, the differences between the two mouse models regarding the liver could be beneficial in the study of the primary and secondary effects associated with ACDase deficiency. While it is difficult to evaluate a single organ in the context of a systemic enzyme deficiency, the *Asah1*^P361R/P361R^ mouse model could be utilized to study FD, in which the liver pathology is more severe, while *Asah1^tmEx1^* has the potential to be a translational instrument for more attenuated cases. Development of organ-specific enzyme function deletions could also be useful here. 

Alsamman et al. assessed the effect of knocking out ACDase in hepatic stellate cells by breeding Eliyahu et al.’s *Asah1* floxed mice to *Pdgfrb*-Cre mice [31,69,70]. PDGFRβ (platelet-derived growth factor receptor beta) upregulation occurred with myofibroblast activation in a broad range of tissues, including hepatic stellate cells [69]. Their resulting *Asah1*^cko^ mice were studied in a carbon tetrachloride (CCl_4_)-induced model of liver fibrosis [69]. Alsamman et al. found reduced liver fibrosis in these *Asah1*^cko^ mice [69]. They attributed this finding to modulation of transcriptional co-activators Yes-associated protein/Transcriptional coactivator with PDZ-binding motif (YAP/TAZ) [69]. YAP/TAZ plays a role in mechanosensing and regulation of fibroblast activation [69]. Hepatic stellate cells isolated from *Asah1*^cko^ mice displayed decreased *Asah1* expression and reduced YAP/TAZ transcriptional target activation in comparison to control mice. YAP/TAZ activity was found to be inhibited due to its increased proteasomal degradation in *Asah1*^cko^ mice [69]. The Alsamman study not only highlights the key mechanistic role of ACDase in liver function and fibrosis, but further exemplifies how the modulation of ACDase may have therapeutic potential [69].

### 2.3. Cardiac

Cardiac involvement in ACDase deficiency is one of the lesser characterized pathologies; however, abnormalities in several FD patients have been noted. The heart is often recorded as an affected organ in case studies of FD—with little added detail [16,71,72]. Early studies presented six FD patients with cardiac involvement, specifically noting one case with a grade 3/6 systolic murmur potentially due to granulomatous lesions of the heart valves [16]. Lesions such as these were published in three of the six patients in that cohort [16]. In a 2020 study of 15 Egyptian FD patients, echocardiographic abnormalities, including thickened mitral valve leaflets, grade I mitral regurgitation, and minimal patent ductus arteriosus, were reported [65]. Additionally, cardiac ultrasonography conducted on a boy with type 4 FD revealed nodules in the mitral valve [55]. Evidence of Cer accumulation, decreased ACDase activity, and histiocytic infiltration has been reported in the hearts of *Asah1*^P361R/P361R^ mice [35,36], while cardiac pathology in the *Asah1^tmEx1^* mouse remains uncharacterized [44]. Thus, cardiac pathology in these mouse models of ACDase deficiency has yet to be thoroughly elucidated. 

### 2.4. Hematologic

Inflammation is a pervasive component of ACDase deficiency, underscoring the critical role of the hematopoietic system in FD and SMA-PME [15,43,44]. The formation of subcutaneous nodules and accumulation of storage material containing foamy histiocytes and macrophages are characteristic of FD. Upon ultrastructural analyses, the presence of various semi-curvilinear inclusions, often referred to as zebra bodies, banana bodies, and Farber bodies, has been observed in hepatocytes, Kupffer cells, histiocytes, fibroblasts, and endothelial cells from patients with FD [46,47,48,49,50]. Subcutaneous nodules, along with histiocytic infiltration, have been reported not only in FD patients’ joints and extremities, but also in the bone marrow, lung, liver, lymph nodes, spleen, thymus, and heart [73,74,75]. These characteristic nodules on various organs have helped diagnose FD when it was otherwise unsuspected as a differential diagnosis [55]. In one case, FD was diagnosed after histiocytes were found in a patient’s bone marrow aspirate [76]. Hematological abnormalities such as low serum iron, anemia, leukocytosis, monocytosis, thrombocytosis, and nucleated red blood cells have all been published in clinical FD case studies [65,74,77,78]. Interestingly, following allogeneic hematopoietic cell transplantation, 10/10 patients with FD had complete and persistent resolution of their inflammatory symptoms [79]. 

In several severe cases of FD, bloodwork revealed an increased erythrocyte sedimentation rate (ESR), plasma chitotriosidase, and C-reactive protein as well, all reflecting inflammation and monocyte/macrophage activation [80,81,82,83]. Splenomegaly was noted in a few patients with FD; however, in the few case studies published, this finding is described less often than hepatomegaly [16]. X-ray findings showed lymphadenopathy and calcification of the axillary lymph nodes in patients with FD [78]. Enlarged lymph nodes have also been reported in post-mortem studies on FD patients [50,52,74]. 

The *Asah1*^P361R/P361R^ mouse displayed signs of leukocytosis with increased total white blood cell counts [35,36]. Specifically, significantly higher levels of monocytes, eosinophils, and neutrophils were observed [35,36]. Interestingly, despite a trend towards increasing numbers, there were no significant differences in lymphocyte and basophil levels [35,36]. *Asah1*^P361R/P361R^ mice exhibited elevated serum MCP-1 levels, which have previously been discussed by our group as a potential biomarker for FD in patients [40]. Hemoglobin and erythrocyte counts were significantly increased above those of heterozygous controls only. Enlargement of hematopoietic organs was also present in *Asah1*^P361R/P361R^ mice [35,36]. Increased organ size was suggested to be due to the steady accumulation of foamy macrophages, which may damage the organ tissue architecture [36]. Similar findings are described in most cases of patients with ACDase deficiency [16]. Furthermore, lymphoid progenitors were reduced over time, and myeloid progenitors were increased in the bone marrow and thymus of *Asah1*^P361R/P361R^ mice [36]. Only 7- and 9-week-old *Asah1*^P361R/P361R^ mice had significantly decreased CD4/CD8 double-positive T cells of the thymus, indicating a progressive decline [36]. It is suggested that both hematopoietic progenitor populations are not altered by Cer accumulation at an intrinsic level, but instead, ACDase deficiency affects the development of hematopoietic progenitor cells [36]. 

In contrast to what was observed in the *Asah1*^P361R/P361R^ mouse, the *Asah1^tmEx1^* mouse displayed decreased circulating B and T lymphocytes in peripheral blood, while numbers of granulocytes and monocytes were not statistically different when compared to those of wild-type control mice [44]. The mean corpuscular volume and average mass of hemoglobin in erythrocytes from peripheral blood of *Asah1^tmEx1^* mice were significantly reduced, while the number of erythrocytes was significantly increased [44]. Hematocrit and platelet counts were normal in the *Asah1^tmEx1^* model [44]. Furthermore, the *Asah1^tmEx1^* mice exhibited splenomegaly, yet with a reduced number of splenocytes [44]. The size of the thymus and lymph nodes was unchanged; however, both organs had significantly reduced cellularity [44]. The architecture of the spleen, lymph nodes, and thymus was altered due to foamy macrophage infiltration, mirroring the findings in most FD patient studies [16]. Inflammatory markers MCP-1, MIP-1α, VEGF, and IP-10 were elevated in the serum of the *Asah1^tmEx1^* mice [44]. Parallel to what was previously described in the *Asah1*^P361R/P361R^ model, CD4/CD8 double-positive T cells of the thymus in *Asah1^tmEx1^* mice were decreased, coupled with a considerable reduction in CD4 and CD8 single-positive T cells [36,44]. Investigation of the bone marrow of *Asah1^tmEx1^* mice uncovered the expansion of the early hematopoietic stem and progenitor cell compartment and the lineage committed progenitor cell compartment [44]. Interestingly, long-term stem cells were unaffected. The increase in the early progenitor cell compartment was confirmed to not induce the higher blood cell counts observed [44]. Despite the extensively reported hematopoietic and storage pathologies, neither the *Asah1*^P361R/P361R^ nor *Asah1^tmEx1^* mouse model was found to develop subcutaneous nodules [35,44]. The highly irregular presentation of hematopoietic consequences in patient cases and the two mouse models of ACDase deficiency exemplifies the variable nature of these disorders (Figure 1).

### 2.5. Neurologic

The effect of ACDase deficiency on the nervous system is extensive, affecting both the central and/or peripheral nervous systems. Neurological effects are seen across the entire spectrum of FD, with some patients experiencing progressive neurological deterioration, seizures, or developmental delays, leading to intellectual disability, and wheelchair dependence [16,51,71,84,85]. In a case of two sisters, both developed progressive loss of speech, ataxia, and paraparesis after the age of 1 years old [16]. Post-mortem analysis revealed that both sisters displayed neuronal loss, histiocytic infiltration, and vacuolization of neuronal cytoplasm [16]. Another FD case recorded a young girl who began to develop ataxia, rigidity, seizures, tremors, polymyoclonia, and dementia at the age of 2.5 years old [16]. In a unique case of FD with Gaucher-like symptoms, a patient presented with decompensated communicating hydrocephalus [86]. The lateral and third ventricles were found to be severely dilated with periventricular hyperintensity on T2 fluid-attenuated inversion recovery magnetic resonance imaging (MRI) images [86]. MRI was also used to show atrophy of the brain, hydrocephalus, and cortical brain in a FD patient who had undergone hematopoietic stem cell transplantation (HSCT) [87,88]. Increased size of the fourth ventricle and pericerebellar sulci were key findings. Upon closer examination, storage pathology was observed in the peripheral nervous system (PNS), in which myelinating and non-myelinating Schwann cells had large, membrane-bound inclusions [87]. Specifically, the storage of hydroxy C24:0 and C22:0 Cers was noted in the cerebellum [88]. In a 9.5-month-old FD patient, spatially restricted neurodegeneration of cortical layers 2 to 3 was present, along with decreased myelin, increased microglial proliferation, and fibrous gliosis [52]. It was speculated that the pathology in the anterior horn cells and peripheral neuropathy are associated with the hypotonia, atrophy, and muscle weakness seen in multiple patient cases of FD [38]. 

SMA-PME is primarily characterized by musculoskeletal and neurological pathology. Chronic demyelinating peripheral neuropathy, scoliosis, and sensorineural hearing loss are commonly noted manifestations of SMA-PME [89]. Frequently, the development of epilepsy follows the onset of neuronal disease in SMA-PME patients, manifesting as myoclonic seizures with a series of proximal upper limb jerks along with mobility impairment, cognitive decline, and difficulty swallowing [26,27,28,72]. Eyelid myoclonic status epilepticus has been observed in a SMA-PME patient as well [72]. 

Interestingly, in 2021, the Sacconi group described a case of an adult FD patient developing multisystemic neurological symptoms beyond lower motor neuron deficits, illustrating the broad spectrum of neurological involvement associated with ACDase deficiency and confirming the first case of SMA progressing into FD [17]. Another recent study included a patient in which symptoms of classical FD and SMA-PME were both present [18]. A 3-year-old boy presented with limited mobility, joint pain, and subcutaneous nodules. Neurological examination revealed tongue fasciculations, weakness, diminished reflexes, and bilateral and lower extremity tremor [18]. The patient did not have seizures, and MRI of his brain and spine was normal [18]. However, electrodiagnostic evidence revealed the presence of chronic motor neuropathy with significant fibrillation potentials and large motor units with mild reductions in recruitment in the tibialis anterior muscle [18]. Reduced peroneal motor amplitude from the deep peroneal motor nerve was observed in this patient, along with normal sural sensory response and normal conduction velocity [18]. In 2016, Filosto et al. published a case of two sisters with mutations in the *ASAH1* gene who presented with an SMA phenotype without PME [72]. Both patients displayed progressive muscle weakness, difficulty walking, and postural tremor with no signs of seizures or myoclonus [72]. This presentation might be a protracted form of SMA; it is unclear whether myoclonic epilepsy would manifest later. These cases highlight the unknown pathogenicity and causality associated with mutations in the *ASAH1* gene and disease presentation [17,18,72]. 

The CNS pathology has been thoroughly characterized in the *Asah1*^P361R/P361R^ mouse [38]. Findings such as hydrocephalus affecting all ventricles, impaired motor coordination, cortical neurodegeneration, and immune cell infiltration overlap with descriptions of FD in patients [38,61,84,90]. Microglial and macrophage pathology is minimally described in FD patient cases; however, granuloma-like accumulations of abnormal CD68+ microglia and/or macrophages are among the primary pathological findings in *Asah1*^P361R/P361R^ mice [38]. Interestingly, the accumulation of these cells was perivascular in white matter, potentially contributing to impaired CNS perfusion. Macrophages, peripheral in origin, were further characterized as disrupting neuronal tracts in the spinal cord, pushing on axons, and altering myelin sheaths in sciatic nerves [35,38]. Cer(d18:1/16:0) and Cer(d18:1/18:0) were abundant in *Asah1*^P361R/P361R^ brain tissue as measured by quantitative liquid chromatography–mass spectrometry (LC-MS) [38]. Cer(d18:1/16:0) was primarily localized in the cerebellar granule cells layer, cerebral cortex, and paraventricular and medial thalamic nuclei but was absent in cerebellar white matter. Cer(d18:1/18:0) was localized to the cerebral cortex, septal nuclei, and corpus callosum, while Cer(d18:0/16:0) was concentrated in the midbrain [38]. Ganglioside and sulfatide species were confined to unique regions of the cerebrum and cerebellum [38]. Zebra-, Farber-, and banana-like bodies were noted in cerebellar white matter and cerebrum, analogous to what is described in patient cases [38,52]. 

*Asah1^tmEx1^* mice exhibited neurogenic atrophy and muscular dystrophy [44]. Creatine kinase and lactate dehydrogenase levels increased over time, revealing overlaps with the myocyte damage seen in SMA-PME patients [44,91]. Storage pathology was also reported in *Asah1^tmEx1^* mice, specifically in the corpus callosum [44]. In contrast to that seen in the *Asah1*^P361R/P361R^ mouse, significant demyelination was not observed [38,44]. Instead, focal infiltration by enlarged macrophages containing cathepsin D immunoreactive lysosomes was shown in the corpus callosum, the subcortical white matter of the hemispheres, the optic tract, the corticospinal tract localized to the brain, the cerebellar white matter, the pyramidal tract, and all spinal tracts [44]. Neurons within the first two cortical layers displayed elevated cathepsin D immunoreactivity. Likewise, the choroid plexus cells revealed a potential hydrocephalus phenotype [44]. Similar to the *Asah1*^P361R/P361R^ mouse, the *Asah1^tmEx1^* mouse also had perivascular macrophage accumulation in the meninges and cerebellum [38,44]. Macrophage infiltrates in the intraparenchymal CNS were present alongside astroglial activation. The olfactory bulb displayed signs of macrophage accumulation as well [44]. In summary, both *Asah1*^P361R/P361R^ and *Asah1^tmEx1^* mice are compelling pre-clinical models for therapy development, as CNS involvement remains a large hurdle in the treatment of ACDase deficiency [15,38,44,92,93,94].

### 2.6. Dermatologic 

Beyond the classic subcutaneous nodules, skin lesions and plaques are frequently mentioned in FD patient cases [77,95,96]. Histiocytic infiltration, storage pathology, hyalinized collagen, and hyperkeratosis have been seen in dermal biopsies from patients with FD [77,97,98]. Farber bodies were present in storage vacuoles and histiocytes, similar to what is observed in other tissues affected by ACDase deficiency [66,96,98]. More rare findings include a case describing an infant with clinical signs comparable to skin stiffness syndrome [99]. The patient presented with thick, hardened skin, a stiff neck, and scleroderma-like patches; he died around the age of 2 years old. Interestingly, a milder case of a 25-year-old female with FD presented with overlapping, redundant skin with a rubber-like texture [100]. That patient’s 29-year-old brother was also mentioned to have developed this “redundant” skin phenotype [100]. Genetic analysis conducted on the Yoruba family in Nigeria revealed that heterozygous *ASAH1* mutations may cause susceptibility to keloid formation [101]. Of 24 family members ranging from ages 2 to 57 years old, nine had keloids in varying locations, and two had hypertrophic scarring. Linkage analysis and exome sequencing led to the identification of an L386P mutation in the *ASAH1* gene responsible for the findings in this family [101]. Although no enzyme activity measurement or lipid analysis was conducted, this novel variant broadened what we know about the dermatological presentation of ACDase deficiency [101]. 

Although the *Asah1*^P361R/P361R^ mice did not develop subcutaneous nodules, as seen in FD patient cases (Figure 1), progressive skin defects coupled with decreased skin elasticity and reduced skin thickness mirrored the case findings of the infant with indurated skin [40,43,99]. *Asah1*^P361R/P361R^ mice also exhibited significant inflammation, complete loss of hypodermal adipose tissue, altered lipid composition in skin lysates, and abnormal Cer accumulation in individual skin cells such as fibroblasts, keratinocytes, and adipocytes [43]. Significant storage of nondegraded substrates and macrophage infiltration affected mainly the dermal reticular layer spreading to the connective tissue of the hypodermis in this model. Moreover, reduced proliferation of *Asah1*^P361R/P361R^ mouse fibroblasts and adipose-derived stem/stromal cells, along with impaired adipogenesis of adipose-derived stem/stromal cells, was also reported [43]. Dermatologic manifestations of ACDase deficiency have not been published in the *Asah1^tmEx1^* model.

### 2.7. Ovarian 

No specific descriptions of an ovarian phenotype have been published in FD patient studies, as many of the patients do not survive past childhood [15]. A singular case described an instance of intrauterine death suggested to be a result of FD [75]. 

The Schuchman group has studied ACDase’s role in reproduction and importance in developmental biology. As previously mentioned, while generating an ACDase-deficient mouse, the group noted that their *Asah1* knockout embryos did not survive past the two-cell stage and, instead, underwent apoptotic death [33]. The authors proposed that ACDase inhibits the default apoptosis pathway in embryos due to its activity on Cer and/or S1P [33]. After generating the TM-induced conditional *Asah1* knockout (cACKO) mouse in 2012, the Schuchman group made several discoveries pertaining to ovarian pathologies in relation to ACDase deficiency [31]. Intraperitoneal delivery of TM in 5-week-old female mice impaired fertility due to inadequate follicle maturation [31]. Apoptosis of oocytes occurred during the progression of follicles from the secondary to antral stages. Specifically, the cACKO mice had decreased antral follicles and ovarian reserves, resulting in the observed reduction of fertility. Increased total Cer levels were present in ovarian connective tissue and early antral follicle theca cells of TM-treated conditional ACDase knockout mice when compared to control mice. Bone-marrow-derived macrophages from ACDase knockout mice exhibited increased Cer(16:0) [31]. Given these findings, the Schuchman group demonstrated key involvement of ACDase in the development of follicles and oocyte survival as well [31]. 

Likewise, examination of *Asah1*^P361R/P361R^ mice also revealed a reduced number of follicles, notably at the antral stage, similar to what was seen in the cACKO mouse [31,35]. The ovaries of the *Asah1*^P361R/P361R^ mice were also physically smaller and had less fat covering [35]. 

### 2.8. Ocular 

A common ocular manifestation reported in FD patents is a grayish retina with a cherry-red spot in the macula [16,52,56,58,102,103]. These patients often have neurological involvement, along with other ocular phenotypes such as nystagmus, corneal and lenticular opacities, macular degeneration, and xanthoma-like growth on the conjunctiva [54,65,84,104]. In one case, a granulomatous lesion of the conjunctiva was reported [16]. In a more recent case, fundus examination of the eye revealed partial optic atrophy in a patient with FD [65]. In the previously discussed case of a patient with Gaucher-like FD, a novel phenotype of bilateral medial pinguecula was revealed upon ocular examination [86]. This patient also exhibited bilateral papilledema, loss of menace reflexes, delayed pupillary light response, and intermittent following and fixation of light [86].

*Asah1*^P361R/P361R^ mice displayed a similar progressive optic nerve and retinal pathology (Figure 1), allowing for a closer examination of ACDase’s effects on the ocular system [41]. In the *Asah1*^P361R/P361R^ mouse, retina physiology and visual function, including retinal thickness, progressive inflammation, neuroaxonal dystrophy, and reduced axonal density, were severely impaired [41]. Visual cliff behavioral studies using *Asah1*^P361R/P361R^ mice revealed dramatically decreased depth perception in these ACDase-deficient mice. C16:0, C18:0, C20:0, C22:0, C24:1, and C24:0 Cers were elevated in the retina of *Asah1*^P361R/P361R^ mice by 8 to 9 weeks of age [41]. Additionally, astrogliosis and storage pathology were noted in the retina, optic nerve head, and unmyelinated optic nerves. Infiltrating macrophages, increased astrocyte counts, and abnormal storage vacuoles containing Farber bodies were also detected in the myelinated optic nerve. The *Asah1*^P361R/P361R^ mouse model suggests that the sphingolipid imbalance caused by ACDase deficiency results in visual impairment, progressive inflammation, and dysmorphic retinal and optic nerve pathology [41]. The only ocular manifestation reported in *Asah1^tmEx1^* mice was “ballooned macrophages containing cathepsin D immunoreactive lysosomes” in the optic tracts [44]. 

### 2.9. Gastrointestinal 

ACDase deficiency and its effects on the GI system have not been thoroughly examined in patients with FD. That said, some gastrointestinal phenotypes have been described in patients with FD or SMA-PME. Symptoms such as persistent diarrhea, gastrointestinal lesions, and erosion of gastrointestinal mucosa have been described [77,105]. Additionally, analysis of biopsied colon tissue from a patient with FD revealed increased apoptosis in crypt cells [106]. Caspase-3-positive cells were found to co-localize with GD3 ganglioside-positive cells [106]. This resulted in speculation that the apoptosis of colon crypt cells may be associated with the synthesis of GD3 as a result of the accumulation of Cers [106]. Selectively elevated ACDase expression in the inflammatory infiltrate in human and murine colitis has been described (Figure 1) [49,107,108]. Minimal work has been conducted assessing the GI tract in ACDase-deficient mouse models. In the myeloid-specific ACDase conditional knockout mouse model Mye AC cKO, Obeid et al. described suppression of intestinal neutrophil recruitment to the colon mucosa because of defective cytokine and chemokine production, illustrating ACDase’s involvement in intestinal inflammation [49]. The gastrointestinal system has not been studied in the *Asah1*^P361R/P361R^ and *Asah1^tmEx1^* models. Including GI studies in these models may help to uncover more details about the manifestations of ACDase deficiency in relation to the GI system.

### 2.10. Renal

Despite kidneys being described as having a high expression of ACDase, little is known about the clinical presentation of renal pathology in ACDase deficiency beyond the increased renal Cer accumulation observed in several patients (Figure 1) [16,109]. In the *Asah1*^P361R/P361R^ mouse, kidneys had elevated Cer levels (Figure 1) [35]. Yet, a detailed histological analysis has not been reported. *Asah1^tmEx1^* mice displayed elevated kidney total Cers yet lacked histological abnormalities [44]. Serum creatinine levels were elevated in *Asah1^tmEx1^* mice when compared to wild-type mice; however, there were no statistical differences [44]. Blood urea nitrogen, however, was elevated in the *Asah1^tmEx1^* mice, indicating a potentially decreased glomerular filtration rate [44]. As previously mentioned, podocyte-specific *Asah1* gene deletion caused glomerular injury characterized by proteinuria and albuminuria in mice [48]. Podocytes of *Asah1*^fl/fl^/Podo^Cre^ mice had foot process effacement and abnormal microvillus formation when compared to controls at 8 weeks [48]. When compared with wild-type and *Asah1*^fl/fl^/WT mice, total Cers and C16:0 were significantly elevated in the glomeruli of *Asah1*^fl/fl^/Podo^Cre^ animals [48]. Interestingly, there were no differences in sphingosine levels between all compared genotypes [48].

### 2.11. Osteological

Paired with the classic subcutaneous nodules, the interphalangeal joints, knees, ankles, and wrists are often deformed, swollen, and painful in FD patients [16,17,18,59,65,86,100,110]. In these cases, worsening osteopenia and osteoporosis are often present in long bones, metatarsals, phalanges, and metacarpals [54,74,77]. Progressively shortening fingers and toes have been described in several cases [100,111]. In addition to shortened fingers and toes, one group reported severe peripheral osteolysis in all three siblings with FD that were studied (Figure 1) [112]. In a recent case of a 23-month-old with FD, scalloping of the distal ulna, bulky hyperemic synovium, and multiple foci of tenosynovitis without joint effusion were discussed [110]. Irregular borders of the distal metaphysis of metacarpals were also noted in addition to enlarged joints and subcutaneous nodules [110]. A case described as FD “mimicking Gaucher disease” presented a patient with Erlenmeyer flask deformity of the femur, a novel finding in FD [86]. Osteologic deformities are present in SMA-PME as well, with scoliosis being the most prevalent [21]. In a 2013 study of a 9-year-old girl described as having FD with PME, tumorous, osseous lesions grew on her spine that resulted in odontoid destruction due to inflammation [113]. After undergoing two HSCTs, her mobility improved, but the myoclonic epilepsy persisted [113]. Another case in which the patient had both FD and SMA-PME, peripheral joint swelling, restricted mobility, and a right hip effusion was noted [18]. Decreased range of motion and pain remain some of the most debilitating components of FD and SMA-PME, with many patients needing a mobility aid. 

Although the bones have not specifically been investigated in *Asah1*^P361R/P361R^ or *Asah1^tmEx1^* mice (Figure 1), both models are smaller in size and weight when compared to wild-type controls, pointing to shorter bone lengths. Interestingly, *Asah1^tmEx1^* mice exhibited SMA-PME-like symptoms of wasting, scoliosis, and a “waddling gait” [44]. Further analysis is needed to draw conclusions regarding osteological parallels between existing mouse models and the human FD and SMA-PME conditions.

## 3. Therapeutic Approaches

### 3.1. Current Treatment

Currently, there is no cure for ACDase deficiency. Treatment approaches focus on symptom management. Corticosteroids and physical therapy are used to ameliorate pain and immobility [71,107,108,114]. In cases of severe nodule formation, surgical intervention is common [115,116]. HSCT is a treatment option for patients with less severe cases of FD (Table 1) [71,92]. Considerable improvements in mobility and pain have been seen post-HSCT in FD patients without CNS involvement [71,92]. In cases of FD patients with CNS manifestations, HSCT was found to elevate ACDase activity in peripheral blood leukocytes, resolve subcutaneous nodules, alleviate pain, and reduce hoarse voice but did not relieve neurological symptoms [71,92,117,118]. Unfortunately, some patients later died of progressive neurological deterioration approximately 28 months after HSCT [118]. In a recent long-term study of 10 FD patients who had undergone HSCT 15 years prior, eight were still alive at the time of publication [79]. The mean survival time was reported to be 10.4 years, with resolved inflammatory joint disease, variable respiratory improvements, and persisting neurological symptoms in the surviving patients [79]. 

For SMA-PME, there are no general guidelines for treatment due to the small number of published cases. Treatment is usually individualized on a case-by-case basis. In addition to general palliative care, patients are usually first treated with antiepileptic drugs to reduce seizures [119]. Physical therapy and psychotherapy are also incorporated into treatment plans [120]. As with FD, SMA-PME patients often suffer severe respiratory complications and require a tracheostomy, mechanical ventilation, and gastric feeding [25,119,120].

### 3.2. Enzyme Replacement Therapy

Enzyme replacement therapy (ERT; or enzyme therapy, ET) is a standard treatment for several LSDs, including but not limited to Fabry disease, Pompe disease, Gaucher disease, mucopolysaccharidosis (MPS) I, MPS II, and MPS VI [121]. ERT using recombinant human ACDase (rhACDase) is currently being studied as a potential treatment for ACDase deficiency and related conditions involving Cer accumulation such as cystic fibrosis (CF) [122]. Encouragingly, rhACDase can be produced and purified on a large scale [123]. Lowered Cer accumulation, decreased MCP-1 expression, and minimized macrophage infiltration were all observed in ERT-treated *Asah1*^P361R/P361R^ mice [123]. These results are promising; however, there are still several limitations that must be addressed (Table 1). Translating the treatment to humans and determining the proper dosage remain key challenges [15]. Additionally, ERT does not yet have the full capacity to penetrate the blood–brain barrier [15]. In more severe cases, CNS involvement remains a large hurdle when treating ACDase deficiency [124]. Localized enzyme administration and fusion proteins targeting the CNS are currently being studied as potential ways to circumnavigate these barriers [125]. Some caution is also warranted concerning the implementation of ERT for FD. ACDase has been found to be upregulated in a number of cancers, including but not limited to: prostate cancer, melanoma, rectal cancer, and acute myeloid leukemia [126,127,128,129]. Thus, animal models allowing selective over-expression of this key regulatory enzyme in specific tissues could be very useful to the field.

### 3.3. Pharmacological Chaperone Therapy

Pharmacological chaperone therapy (PCT) is a treatment option that targets molecular chaperones and related folding factors in LSDs [130]. Pharmacological chaperones are small ligands engineered to selectively bind and stabilize defective (but not catalytically inactive) enzymes to improve function and lysosomal trafficking. PCT has been proposed for LSDs such as Fabry and Gaucher disease, in which hydrolase misfolding causes pronounced effects [130]. PCT shows promise in addressing certain limitations of other existing therapies as well (Table 1). As small molecules, pharmacological chaperones provide oral bioavailability, extensive tissue distribution, and cell membrane diffusion [130]. A major limitation of this therapy, however, is that chaperones are usually reversible, competitive inhibitors of the target enzymes, often not offering enough benefit [130]. Furthermore, PCT is not always applicable to all mutations and commonly is only useful for missense mutations in which the entire enzyme is not lost or destroyed [121,130]. Therapeutic chaperones have been found to be most efficient when there is residual enzyme activity [130]. For these reasons, chaperone response is heavily dependent on the type of mutation and the specific chaperone molecule involved [130]. Further study, along with advances in individualized screening strategies, is needed to identify whether FD and SMA-PME are good candidates for PCT.

### 3.4. Gene Therapy

Gene therapy is being investigated for the treatment of several LSDs. A recent clinical trial headed by our group using lentiviral (LV) vector-based gene therapy to treat the monogenic LSD Fabry disease demonstrated compelling results with the vector and enzyme detected long term in peripheral blood (ClinicalTrials.gov NCT02800070) [131]. Bone marrow cells, leukocytes, and plasma demonstrated enzyme activity within or above the reference range [131]. Plasma and urine globotriaosylceramide and globotriaosylsphingosine were reduced [131]. Ex vivo gene therapy has the potential to correct neurological involvement (Table 1), as seen in LV HSC-based gene therapy targeting the LSD metachromatic leukodystrophy (ClinicalTrials.gov NCT01560182) [132]. In that trial, all patients had polyclonal hematopoietic reconstitution of gene-corrected cells without signs of genotoxicity [132]. Furthermore, all patients were described as having nearly complete gene-marked hematopoiesis [132]. Most of the treated patients displayed normal cognitive development in addition to decreased demyelination and brain atrophy throughout follow up [132].

Gene therapy for ACDase deficiency is yet to be conducted in clinical trials. Like other monogenic LSDs, ACDase deficiency is caused by a defect in a single gene, *ASAH1*, making it an attractive candidate for gene therapy. In 1999, an early ‘proof-of-concept’ study conducted by Medin et al. utilized a recombinant oncoretrovirus encoding the human ACDase cDNA sequence [133]. Re-establishment of enzymatic activity and increased Cer catabolism were reported in transduced fibroblasts from FD patients. Spent culture media containing secreted ACDase from the transduced cells were able to cross-correct other untreated bystander cells. Upon closer examination, it was found that the non-transduced cells took up functional ACDase from their environment through the mannose-6-phosphate receptor pathway [133]. Transducing human ACDase cDNA into murine CD34+ stem/progenitor cells and non-human primate analogous cells was also successful [134]. Elevated ACDase activity and decreased Cer were recorded in various blood cells more than one year post treatment [134]. Using recombinant oncoretroviral and LV vectors, a more recent study from the Medin group reported similar gene correction effects with elevated ACDase activity and restored Cer catabolism in human HSCs [92]. Directly injecting the vector into murine neonates was also successful, resulting in ACDase expression for at least 13 weeks [93]. A similar method was later used in *Asah1*^P361R/P361R^ mice, increasing lifespan from 9–10 weeks to 16.5 weeks [35]. Given the above, ex vivo transduction of patient-derived stem cells followed by HSCT may be a strategy for long-term treatment of ACDase deficiency. 

In 2022, Zhang et al. published the first study of organ-specific gene augmentation therapy for the treatment of FD (Table 1), in which recombinant adeno-associated virus (rAAV) vector-mediated over-expression of ACDase ameliorated retinopathy in *Asah1*^P361R/P361R^ mice [135]. Treated *Asah1*^P361R/P361R^ mice had increased protection against the ocular disease manifestations associated with ACDase deficiency such as abnormal retinal morphology, increased retinal thickening, inflammation, and pathological reflectivity [135]. Importantly, treated *Asah1^+/P361R^* and *Asah1^+/+^* mice displayed opposite results, indicating that over-expression of ACDase can be deleterious and sometimes unpredictable [135]. This study highlighted the potential of ACDase gene therapy for the treatment of FD [135].

Several clinical studies using gene therapy to treat SMA1 have been conducted. Using a non-integrating adeno-associated virus serotype 9 (AAV9) encoding human survival of motor neuron (SMN1) cDNA, a phase 1/2 trial opened in 2014 and was conducted in 15 patients with SMA type I [136,137]. All patients in the experimental group showed signs of maintained or improved respiratory status and increased mobility [136,137]. Several follow-up phase 3 studies have since been conducted, with positive results (ClinicalTrials.gov NCT03306277, NCT03837184, NCT03461289, NCT03837184, NCT03505099, NCT03421977, and NCT04851873), and the gene therapy was approved in the U.S. by the FDA in 2019 and in Europe by the EMA in 2020 [138]. The success of SMA1 gene therapy provides an indirect rationale for the potential use of gene therapy for SMA-PME. 

## 4. Future Directions

Despite recent advances, many uncertainties remain in terms of mechanism, diagnostics, and treatment of ACDase deficiency. Research continues to describe novel, expansive roles demonstrated by ACDase, further revealing its complexity. Studies in the last year have presented ACDase as a target for cancer therapy [10,11,12]. Inhibition of ACDase in melanoma, glioblastoma, and pancreatic cancer, among others, has been shown to have antitumor effects [10,11,12,139]. For example, melanoma cells with *ASAH1* ablated via CRISPR-Cas9 gene editing lost the ability to self-renew and create cancer-initiating cells, highlighting ACDase’s role in autophagy, mitochondrial homeostasis, and maintaining malignancy [10]. Comparable results were seen in pancreatic tumor cells, where ACDase knockdown with siRNA induced ROS accumulation, mitochondrial dysfunction, and increased apoptosis [11]. When targeted in glioblastoma cells, ACDase inhibition with LV vector-delivered shRNA or carmofur, an inhibitor of ACDase that has been used as an antineoplastic drug since the early 1980s, caused a decrease in AKT signaling, affecting migration of glioblastoma cells [12,140]. Although preliminary, these findings all highlight promising future therapeutic strategies involving ACDase for cancer. 

Contrastingly, ACDase upregulation has also shown potential therapeutic benefits in respiratory illnesses such CF and chronic obstructive pulmonary disease (COPD) [122,141,142,143]. Analysis of lung samples from patients with COPD has shown increased levels of Cers, indicating the potential downregulation of ACDase [144]. ACDase expression is significantly reduced in lung and liquid airway cultures of CF patients [144]. This same finding was observed in the analogous tracheal and bronchial epithelial cells of a mouse model of CF [142]. Early studies proved that inhalation of recombinant hACDase normalizes the levels of sphingosine, Cer, and β1-integrin in CF epithelial cells of mice [145]. Importantly, ACDase displays protective effects against *P. aeruginosa* infection, a main culprit in the pathogenesis of CF [146]. In a closer look at ACDase’s involvement in CF, Becker et al. reported that endogenous over-expression of ACDase in mice with CF exerts a protective effect against bacterial infections and normalizes sphingosine levels in respiratory epithelial cells [142]. These studies emphasize the importance of further studying the mechanistic characteristics of ACDase deficiency and its potential involvement in diseases beyond FD and SMA-PME. 

As animal models continue to provide novel insights into ACDase and Cer biology, there are several organs yet to be characterized in both *Asah1*^P361R/P361R^ and *Asah1^tmEx1^* ACDase-deficient mice. To complete a more thorough picture of ACDase-deficient mouse model pathology, organs such as the heart and kidney should be looked at more closely. The prominent differences in the hematopoietic systems of the two mouse models support the need for thorough characterization in each. These variations may extend beyond the hematopoietic system and aid the development of therapies for the wide spectrum of ACDase deficiency presentations. Fully characterizing a mouse model of SMA-PME, such as the one we have generated (manuscript submitted), will also prove beneficial, as the scarcity of these patient descriptions is even greater than that of patient descriptions of FD. 

Most importantly, novel mechanistic insights are needed surrounding ACDase’s pathogenicity in FD and SMA-PME. Although it is known that ACDase deficiency involves lysosomal dysfunction and Cer accumulation, the specific mechanisms inducing cellular dysfunction and, ultimately, symptoms are yet to be understood. Examining the autophagy lysosome pathway could be key to uncovering the pathogenicity of ACDase deficiency. Lysosomal dysfunction due to disrupted digestion, Cer-accumulation-mediated autophagy suppression, and decreased saposin-D-driven presentation of Cers to ACDase are some hypothesized explanations [10]. However, there are several other questions that remain unanswered; the causality of each variable remains unknown. Experiments studying ACDase’s role in development are needed to distinguish whether the expansive signs and symptoms surrounding ACDase deficiency are due to the accumulation of substrate as the disease progresses or are induced early in development and exacerbated by lysosomal dysfunction. Additionally, the Cer, macrophage, and MCP-1 axis must be thoroughly studied to help advance treatment modalities [35]. In 2021, Brooks et al. created a FD human induced pluripotent stem cell (iPSC) line, TRNDi030-A [147]. TRNDi030-A was generated from the fibroblasts of a male patient with a homozygous variant in the second exon of *ASAH1* [147]. The result was a cell line with a mutant ACDase α subunit, leading to ACDase deficiency. This cell-based model may be of great use for future ACDase-deficiency-related therapeutic and mechanistic studies. Mouse models and this novel FD iPSC line will be key tools in furthering our understanding of ACDase’s mechanistic role in ACDase deficiency.

## 5. Conclusions

ACDase deficiency encompasses a spectrum of disorders including FD and SMA-PME. Due to a wide range of clinical presentations and its rarity, ACDase deficiency is often misdiagnosed as more well-known diseases such as juvenile idiopathic arthritis or pediatric hepatomegaly [55,59,80,83,110]. The rapid progression of FD and SMA-PME makes initial diagnosis crucial for effective treatment and management. Animal models serve as a vehicle for the study of pathological hallmarks of diseases, therapy, and treatment development. We have provided an overview of clinical case descriptions and relevant findings in animal models of ACDase deficiency, commenting on significant overlaps and deficits. Continued study of these models will be a useful resource for improving the understanding of Cer accumulation due to ACDase deficiency, allowing for more patients to be properly identified, thoroughly characterized, successfully diagnosed, and efficiently treated.

## Figures and Tables

**Figure 1 biomolecules-13-00274-f001:**
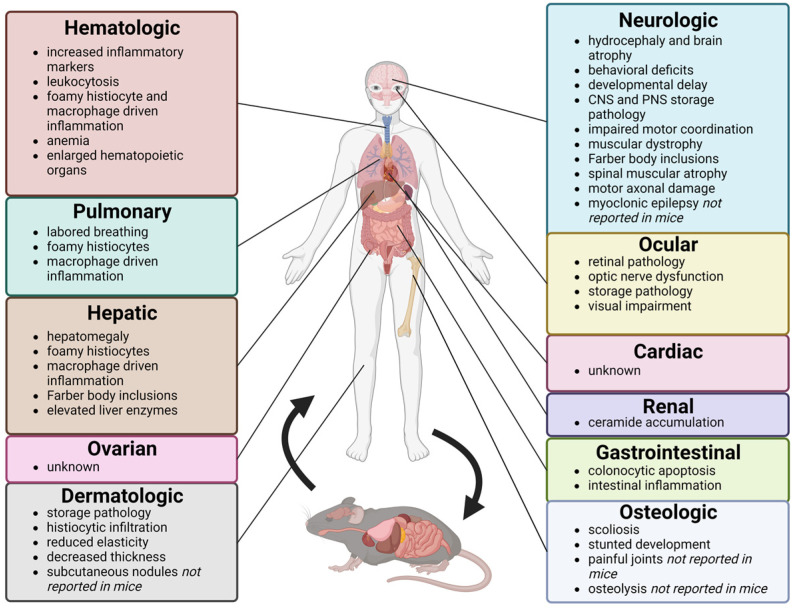
ACDase-deficiency-related disease manifestations reported in both patients and mouse models.

**Table 1 biomolecules-13-00274-t001:** Comparing and contrasting current therapeutic potentials for ACDase deficiency.

Therapy	Benefits	Limitations
Gene Augmentation	Efficiency CNS penetration potential Organ-specific augmentation potential	Variant protein expression remains Dependent on vector and injection site Risks with conditioning
Gene Editing	True correction of mutation CNS correction potential	Less efficiency and higher cost Increased complexity of delivery Off-target effects
Enzyme Replacement	Localized administration potential Fewer conditioning related risks	Unknown dosing Does not penetrate CNS
Pharmalogical Chaperone	Enzymatic correction Oral availability Systemic tissue distribution Cell membrane diffusion	Reversible competitive inhibitors Selective mutation targets
Hematopoietic Stem Cell Transplant	Accessible Well studied	Does not penetrate CNS Risks with conditioning

## Data Availability

Not applicable.

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
