# Peer review of "Acid Ceramidase Deficiency: Bridging Gaps between Clinical Presentation, Mouse Models, and Future Therapeutic Interventions"

_biomolecules, 2023, doi:10.3390/biom13020274_

Round 1
Reviewer 1 Report
Acid Ceramidase Deficiency: Bridging Gaps between Clinical Presentation, Mouse Models, and Future Therapeutic Interventions
This work is well organized, has an extensive literature review, and has successfully filled a breach between clinical knowledge and mouse models of the disease—necessary knowledge to facilitate future research on treatment development.
Some minor mistakes should be corrected.
Line 124-124: "Further analysis identified the mutation responsible within the SMA1 or SMD2 genes". These are not genes, and the sentence does not follow the previous text's idea. The whole sentence is a mistake.
Line 904-907 These are both the same reference (reference # 21 and 25)
Figure 1, "ACDase deficiency-related disease manifestations reported in both patients and mouse models,"
There is no reference in the figure to an important neurologic manifestation: spinal muscular atrophy or motor axonal damage. e, SMA-PME is an important phenotype and should not be neglected.
As a general comment, the recent work from Brooks et al. doi:10.1016/j.scr.2021.102387 on a new Farber disease iPSC line will be worth adding to the review as another valuable resource for studying disease pathophysiology and developing future treatments.
Author Response
This work is well organized, has an extensive literature review, and has successfully filled a breach between clinical knowledge and mouse models of the disease—necessary knowledge to facilitate future research on treatment development. 
Thank you for the positive review of our manuscript. We have incorporated your suggested modifications into our revised document. Point by point replies to your insightful suggestions follow.
Some minor mistakes should be corrected.
Point 1: Line 124-124: "Further analysis identified the mutation responsible within the SMA1 or SMD2 genes". These are not genes, and the sentence does not follow the previous text's idea. The whole sentence is a mistake.
Response 1: Thank you for bringing this to our attention. As you suggested, this sentence needed to be re-written. We have modified the sentence to read: "Further analysis of several patients with SMA-PME identified the mutations responsible within the ASAH1 gene."
Point 2: Line 904-907 These are both the same reference (reference #  21 and 25)  
Response 2: Thank you for catching this duplication. We have removed the duplicated reference and modified our reference numbers throughout in accordance with this change.
Point 3: Figure 1, "ACDase deficiency-related disease manifestations reported in both patients and mouse models,"  
There is no reference in the figure to an important neurologic manifestation:  spinal muscular atrophy or motor axonal damage. e, SMA-PME is an important phenotype and should not be neglected.
Response 3: Thank you for this suggestion. We have modified Figure 1 to incorporate SMA-PME.
Point 4: As a general comment, the recent work from Brooks et al. doi:10.1016/j.scr.2021.102387   on a new Farber disease iPSC line will be worth adding to the review as another valuable resource for studying disease pathophysiology and developing future treatments.
Response 4: Thank you for this suggestion. We have incorporated the recent work from Brooks et al. into the manuscript as follows: “In 2021, Brooks et al. created a FD human induced pluripotent stem cell (iPSC) line, TRNDi030-A (149). TRNDi030-A was generated from fibroblasts of a male patient with a homozygous variant in the second exon of ASAH1 (149). The result is a cell line with a mutant ACDase α subunit, leading to ACDase deficiency. This cell-based model may be of great use for future ACDase deficiency-related therapeutic and mechanistic studies. Mouse models and this novel FD iPSC line will be key tools in furthering our understanding of ACDase’s mechanistic role in ACDase deficiency.”
Reviewer 2 Report
This review by Kleynerman et al is an excellent summary of the current knowledge of acid ceramidase deficiency disorders due to ASAH1 gene mutations that cause Farber disease (FD) and spinal muscular atrophy with progressive myoclonic epilepsy (SMA-PME). The review covers clinical presentations, mouse models, and future therapeutic interventions of acid ceramidase deficiency disorders.
Because of the rarity of FD and SMA-PME, only a limited number of researchers are aware of these diseases. This review will be attractive to many researchers who are eager to gain new knowledge. However, the review not only satisfies some researchers’ desire for knowledge, but is useful in a practical manner for neonatologists, pediatricians and pediatric neurologists treating sick infants with poor activity and/or respiratory problems.
This review may be timely and important because of growing interest in 5q-spinal muscular atrophy (5q-SMA). Recently, 5q-SMA has become a major concern for clinicians treating infants with poor activity and/or respiratory problems because they are required to diagnose 5q-SMA and treat them with new effective drugs at the earliest stage. Any disorders with SMA-like symptoms, including SMA-PME, should be distinguished. Although SMA-PME is a very rare disease, it should always be considered a differential disorder when diagnosing 5q-SMA.
FD is also a poorly recognized disease, and its diagnosis is difficult. Systemic symptoms of FD appear in infants. As mentioned by the authors, the characteristic nodules on various organs may help diagnose FD. This review will be important for clinicians to obtain knowledge regarding FD, making them able to diagnose it, or at least suspect it.
In conclusion, I believe that this review is not only interesting, but useful, regarding real-world medicine.
Minor comments
The review comprehensively describes FD and SMA-PME. However, I have some minor comments, especially regarding the references section.
Line 869
“Acknowledgments: Figure 2 was created with BioRender.com.”
Only Figure 1 and Table 1 are present in the manuscript.
Reference 12
Cells 11, 1873 (2022).
Reference 16
The year of being published might be 2019.
https://ommbid.mhmedical.com/content.aspx?sectionid=225545527&bookid=2709
References 21 and 25 appear to be the same. If these references are duplicated, please renumber the references.
Reference 105
This reference seems to be a supplement. The title is “Farber Lipogranulomatosis: A Report of a Case with Nystagmus, Myoclonus and Convulsions.”
https://www.brainanddevelopment.com/issue/S0387-7604(87)X8018-3
https://www.brainanddevelopment.com/article/S0387-7604(87)80038-4/pdf
https://doi.org/10.1016/S0387-7604(87)80038-4
Author Response
Point 1: This review by Kleynerman et al is an excellent summary of the current knowledge of acid ceramidase deficiency disorders due to ASAH1 gene mutations that cause Farber disease (FD) and spinal muscular atrophy with progressive myoclonic epilepsy (SMA-PME). The review covers clinical presentations, mouse models, and future therapeutic interventions of acid ceramidase deficiency disorders.
Because of the rarity of FD and SMA-PME, only a limited number of researchers are aware of these diseases. This review will be attractive to many researchers who are eager to gain new knowledge. However, the review not only satisfies some researchers’ desire for knowledge, but is useful in a practical manner for neonatologists, pediatricians and pediatric neurologists treating sick infants with poor activity and/or respiratory problems.
This review may be timely and important because of growing interest in 5q-spinal muscular atrophy (5q-SMA). Recently, 5q-SMA has become a major concern for clinicians treating infants with poor activity and/or respiratory problems because they are required to diagnose 5q-SMA and treat them with new effective drugs at the earliest stage. Any disorders with SMA-like symptoms, including SMA-PME, should be distinguished. Although SMA-PME is a very rare disease, it should always be considered a differential disorder when diagnosing 5q-SMA.
FD is also a poorly recognized disease, and its diagnosis is difficult. Systemic symptoms of FD appear in infants. As mentioned by the authors, the characteristic nodules on various organs may help diagnose FD. This review will be important for clinicians to obtain knowledge regarding FD, making them able to diagnose it, or at least suspect it.
In conclusion, I believe that this review is not only interesting, but useful, regarding real-world medicine.
Response 1: We thank the reviewer for their positive review.
Minor comments
Point 2: The review comprehensively describes FD and SMA-PME. However, I have some minor comments, especially regarding the references section.
Response 2: Thank you for so diligently combing our references. We have made the required modifications as listed below.
 
Point 3: Line 869
“Acknowledgments: Figure 2 was created with BioRender.com.”
Only Figure 1 and Table 1 are present in the manuscript.
Response 3: Thank you for bringing this to our attention. We have modified the text to correctly state that this was as you suggested, Figure 1.
Point 4: Reference 12
Cells 11, 1873 (2022).
Response 4: Thank you for bringing this to our attention. We have modified the text to correctly list this reference as follows:. C. C. Hawkins et al., Targeting Acid Ceramidase Inhibits Glioblastoma Cell Migration through Decreased AKT Signaling. Cells 11, 1873 (2022).
Point 5: Reference 16
The year of being published might be 2019.
https://ommbid.mhmedical.com/content.aspx?sectionid=225545527&bookid=2709
Response 5: Thank you for bringing this to our attention. We have modified the text to correctly list this reference as follows: T. Levade, K. Sandhoff, H. Schulze, J. Medin, Acid ceramidase deficiency: Farber lipogranulomatosis. Online Metabolic & Molecular Bases of Inherited Disease (OMMBID). McGraw-Hill, (2019).
Point 6: References 21 and 25 appear to be the same. If these references are duplicated, please renumber the references.
Response 6: Thank you for catching this duplication. We have removed the duplicated reference and modified our reference numbers throughout in accordance with this change.  
Point 7: Reference 105
This reference seems to be a supplement. The title is “Farber Lipogranulomatosis: A Report of a Case with Nystagmus, Myoclonus and Convulsions.”
https://www.brainanddevelopment.com/issue/S0387-7604(87)X8018-3
https://www.brainanddevelopment.com/article/S0387-7604(87)80038-4/pdf
https://doi.org/10.1016/S0387-7604(87)80038-4
Response 7: Thank you for bringing this to our attention. We have modified the text to correctly list this reference as follows: M. Ohfu et al., Farber Lipogranulomatosis: A Report of a Case with Nystagmus, Myoclonus and Convulsions. Brain and Development 9, 227 (1987).